# Evaluation of Virucidal Quantitative Carrier Test towards Bovine Viruses for Surface Disinfectants While Simulating Practical Usage on Livestock Farms

**DOI:** 10.3390/microorganisms10071320

**Published:** 2022-06-30

**Authors:** Md. Amirul Hasan, Yu Miyaoka, Md. Humayun Kabir, Chisaki Kadota, Hakimullah Hakim, Dany Shoham, Harumi Murakami, Kazuaki Takehara

**Affiliations:** 1Laboratory of Animal Health, Cooperative Division of Veterinary Sciences, Graduate School of Agriculture, Tokyo University of Agriculture and Technology, 3-5-8, Saiwai-cho, Fuchu-shi, Tokyo 183-8509, Japan; s194497y@st.go.tuat.ac.jp (M.A.H.); s205896s@st.go.tuat.ac.jp (Y.M.); s184195q@st.go.tuat.ac.jp (M.H.K.); hakim.856@gmail.com (H.H.); murakamh@cc.tuat.ac.jp (H.M.); 2Laboratory of Animal Health, Department of Veterinary Medicine, Faculty of Agriculture, Tokyo University of Agriculture and Technology, 3-5-8, Saiwai-cho, Fuchu-shi, Tokyo 183-8509, Japan; s183553r@st.go.tuat.ac.jp; 3Bar-Ilan University, Begin-Sadat Center for Strategic Studies, Ramat Gan 5290002, Israel; shoham_d@013net.net

**Keywords:** biosecurity enhancement, carrier test, coronavirus, evaluation of disinfectants, food additive grade calcium hydroxide, quaternary ammonium compound, rotavirus, spectral broadening effect, suspension test, synergistic effect

## Abstract

Livestock farming is affected by the occurrence of infectious diseases, but outbreaks can be prevented by effective cleaning and disinfection along with proper farm management. In the present study, bovine coronavirus (BCoV) and bovine rotavirus A (RVA) were inactivated using food additive-grade calcium hydroxide (FdCa(OH)_2_) solution, quaternary ammonium compound (QAC) and their mixture through suspension tests as the primary screening, and afterward via carrier tests using dropping or dipping techniques as the secondary screenings. Viruses in the aqueous phase can be easily inactivated in the suspension tests, but once attached to the materials, they can become resistant to disinfectants, and require longer times to be inactivated. This highlights the importance of thorough cleaning with detergent before disinfection, and keeping elevated contact durations of proper disinfectants to reduce viral contamination and decrease infectious diseases incidence in farms. It was also reaffirmed that the suspension and carrier tests are necessary to evaluate disinfectants and thus determine their actual use. Particularly, the mixture of QAC and FdCa(OH)_2_ was found to exhibit synergistic and broad-spectrum effects compared to their use alone, and is now recommended for use on livestock farms.

## 1. Introduction

Disinfection is an important strategy to reduce the amount of pathogens and the risk of contaminating environments, and disinfectants are important tools for enhancement of livestock farm biosecurity [1,2,3].

In livestock farms, shoe soles and clothing of farm workers can serve as carriers for pathogens. Disinfection of porous and nonporous surfaces is especially important because plastics, rubber, and clothing can be a source of infection for as long as 4 days after the pathogen is attached [1,4]. In addition, environmental factors that can reduce the effectiveness of disinfection include temperature, organic load, and short contact time, which should be considered during disinfection [5,6,7]. In order to prevent livestock farms from being contaminated and to improve biosecurity, it is worthwhile to accurately determine the concentration of the germicidal or virucidal agents and the duration of disinfection to inactivate pathogens. Accurate evaluation methods for surface disinfection are hence required to effectively apply the disinfectants on the farms and improve biosecurity level. 

Enteric viruses constitute an important and also complex community of viruses. Infections with enteric viruses can result in the excretion of large amounts of viral particles in body secretions, particularly in feces; the viral particles are extremely stable in the environment, and are spread through fecal–oral route [8]. Bovine rotavirus A (RVA) and bovine coronavirus (BCoV) are the most common viral pathogens responsible for neonatal diarrhea in calves and result in serious economic consequences in bovine industry [9,10]. The double icosahedral shell of non-enveloped viruses such as RVA is more resilient than bacterial cell membranes or lipid-enveloped viruses, and are likely to be more resistant to disinfection than other farm pathogens [11]. Viral lipid envelopes, especially those found in coronaviruses including BCoV and severe acute respiratory syndrome coronavirus 2 (SARS-CoV-2), can remain viable and infectious, surviving for hours in aerosols and even for days on surfaces [4]. 

Food additive grade calcium hydroxide (FdCa(OH)_2_) is relatively novel compared to other materials that can inactivate pathogens [12,13,14]. Quaternary ammonium compounds (QACs) are commonly used at livestock farms, but their effectiveness is reduced by organic contamination and low temperatures [15]. However, QAC disinfection efficacy is synergistically enhanced even at low temperatures and in the presence of organic matter by adding FdCa(OH)_2_ [16,17]. The synergism may be explained by Ca(OH)_2_’s alkalinity that destructs cytoplasmic phospholipid membranes, thereby affecting the integrity of the cytoplasmic membrane [1,17]. Nonetheless, it should be noted that even with the QAC and FdCa(OH)_2_ mixture, it took over 3 min to inactivate pathogens on abiotic carriers (rubber, plastic or steel) [1,18]. Therefore, our recommendation was that boots should be changed and kept in the QAC and FdCa(OH)_2_ mixture for more than 3 min, since the mixture may inactivate both nonenveloped and enveloped viruses [19].

Changing personal protective equipment (clothing and footwear) could prevent transmission of porcine epidemic diarrhea (PED) virus to sentinel pigs as a result of biosecurity procedures [20]. In September 2017, improved hygiene standards on boots in a bovine operation (farm A) in Ibaraki, Japan resulted in a significant reduction in both calf mortality and detection of four viral pathogen indicators for two years, except for the detection rate of bovine RVA in calves less than 3 weeks old [19,21]. In December 2019, the protocols for boots hygiene were revised on farm A (Figure 1). The farm staff were required to stand on a board and remove their outdoor boots, then wear the boots designated for use inside the calf shed at the entrance of each calf shed. Afterwards, the shed-exclusive boots were obligatorily washed and disinfected in a footbath containing the QAC and FdCa(OH)_2_ mixture until their next use (Figure 1).

Virucidal activity should be tested using both suspension and carrier methods, based on established international standardization principles. In the present paper, this pertains to disinfectants that meet the suspension and carrier test standards described in the American Society for Testing and Materials (ASTM) international standards E-1052-20, E1053-20 and E2966-14 (2019) [22,23,24] and European Union standards [25]. 

In the present experiments, contaminated suspension tests and carrier tests were conducted with 5% fetal bovine serum (FBS) or feces to compare virucidal activities of disinfectants, while simulating dirty conditions. In the carrier tests, rayon sheets were used instead of clothes or coveralls and rubber carriers instead of boots; RVA and BCoV were deposited on those carriers, and then the effects of the disinfectants on the viruses on the carriers were compared by the two following methods: (1) 500 μL of each disinfectant was dropped onto the rayon sheets or the rayon sheets were soaked in a tube containing 400 μL of each disinfectant; (2) 500 μL of each disinfectant was dropped onto the virus on the rubber carrier. 

## 2. Materials and Methods

### 2.1. Test Viruses and Cultures

The RVA strain RVA/Bovine-tc/JPN/AH1041/2019/G8P [1] was isolated from calf feces according to the methods previously described [26], with modifications, and was identified by sequencing (accession number LC656030). The RVA strain was propagated in African green monkey kidney (MA104) cells maintained by Eagle’s minimal essential medium (EMEM) (Nissui, Tokyo, Japan) containing 10 µg/mL trypsin from bovine pancreas (Sigma Chemicals, St. Louis, MO, USA). BCoV strain AH1409 was isolated from calf feces and confirmed by sequencing (accession number LC666740), according to the methods previously described [27] in human rectal adenocarcinoma (HRT-18) cells. HRT-18 cells were kindly supplied from the National Institute of Animal Health (NIAH) (Ibaraki, Japan) and maintained and propagated in Dulbecco’s modified Eagle’s medium (DMEM) (Nissui, Tokyo, Japan). Each virus culture was harvested after the appropriate number of incubation days, centrifuged at 1750× *g* for 15 min to remove cellular debris, and the supernatants were collected, aliquoted and stored at −80 °C until used. The virus stock titers of RVA and BCoV were around 10^7^ plaque forming units (PFU)/mL and around 10^7^ fifty percent tissue culture infectious dose (TCID_50_)/mL, respectively.

### 2.2. Virucidal Agents and Blocking Solution

The QAC (Rontect^®^) was purchased from Scientific Feed Laboratory Co., Ltd. (Tokyo, Japan) and diluted 1:500 (QACx500) with redistilled water (dW_2_), to obtain a final concentration of 200 ppm didecyl-dimethylammonium chloride (DDAC), as recommended by the manufacturer. FdCa(OH)_2_ powder at pH 13 with an average diameter of 10 μm of powder particles [14] was purchased from Fine Co., Ltd. (Tokyo, Japan) and used as 0.17% FdCa(OH)_2_ solution or as a QAC and FdCa(OH)_2_ mixture (Mix500) [16,17].

A blocking solution (30% FBS and 0.7 M HEPES, pH 7.2) was used for stopping virus inactivation reactions, as previously described [1,4].

### 2.3. Experimental Designs

Disinfectant procedures were conducted according to ASTM-E1052, E1053 and E2966 standards with some modifications.

#### 2.3.1. Experiment 1: Suspension Tests for Evaluating the Virucidal Activities of the Solutions against RVA and BCoV

Four hundred microliters of QACx500, 0.17% FdCa(OH)_2_, or Mix500 were mixed with 100 μL of RVA or BCoV in a microtube and incubated for 0 s, 5 s or 1 min at room temperature (RT: 25 ± 2 °C). The virus inactivation was immediately stopped by adding 500 μL of the blocking solution [1,4]. To ascertain the effect of the blocking solution, the test solutions were mixed with the blocking solution before the addition of the virus (considered as 0 s treatment and contact time). For the positive control, 400 μL of dW_2_ was used instead of disinfectants. Then, the remaining virus was titrated. To evaluate the inactivating activity of the test solutions in the presence of organic materials mimicking virus environment on farms, FBS was added to each virus to reach 5%.

#### 2.3.2. Experiment 2: Evaluation of Virucidal Activities of the Test Solutions toward Viruses on Rayon Sheets by the Dipping Techniques

The experiment was designed for field application for farm personnel clothes hygiene. Rayon sheets (double-fold, size 2.0 cm × 2.0 cm, Alphase^®^ 5, Iwatsuki Co. Ltd., Tokyo, Japan) were placed in 1.5 mL microtubes and autoclaved at 121 °C for 15 min. To examine virucidal activity toward viruses on clothes, 100 μL of virus containing 5% FBS was inoculated onto the rayon sheets in microtubes and allowed to dry at RT in a biosafety cabinet for 30 min. After drying, 400 μL of Mix500 or dW_2_ as positive control was added by complete dipping and incubated for 0, 5, 15 and 30 s, respectively. Following incubation times, the treatment of samples was stopped by adding blocking solution (Figure 2). Then, the remaining virus was titrated after making serial 10-fold dilutions.

#### 2.3.3. Experiment 3: Evaluation of Virucidal Activities of Viruses on Rayon Sheets by the Dropping Techniques

Carrier test on rayon sheets with the dropping techniques was performed at RT as described previously [28]. The experiment was designed for field application of disinfectant spray. Rayon sheets (5.0 cm × 5.0 cm) folded into four pieces (2.5 cm × 2.5 cm) were used for dropping carrier test. To examine virucidal activity on clothes, 100 μL of viral solutions containing 5% FBS were spotted onto the rayon sheets and let dry for 30 min in a biosafety cabinet at RT. On each carrier, 500 μL of Mix500 was dropped to cover all spotted areas, and incubated for 10 min or 1 h, respectively. Then, the rayon sheets were placed into stomacher bags (size 100 mm × 150 mm × 0.09 mm, capacity 80 mL; Organo Corporation, Tokyo, Japan) containing 2 mL of the blocking solution to stop the treatment. Rayon sheets were treated with a BagMixer (MiniMix 100 W CC, Practical Japan Inc., Chiba, Japan) to remove viruses from the sheets (Figure 3). Then, the remaining viable virus was titrated.

#### 2.3.4. Experiment 4. Evaluation of Virucidal Activities of Viruses on Rubber Surfaces Contaminated with Feces

Carrier tests of rubber with the dropping technique were performed at RT as described previously [1]. The experiment was designed for field application for boot hygiene. The autoclaved calf feces was ground by mortar. In brief, an amount of 100 μL of viruses containing 5% feces and simulating dirty conditions was spotted on a rubber carrier coupon (around 5.0 cm × 5.0 cm) as described previously [1]. After being air-dried for 60 min, 500 μL of each solution including dW_2_ as the positive control was dropped on each carrier and incubated for 15 s, 30 s, or 1, 3, 5 and 10 min. Then, the virucidal activities of the test solutions were blocked by placing the carriers into stomacher bags containing 2 mL of the blocking solution; dislodging the virus from the carrier surfaces into fluids (Figure 4) and titration (as described previously [1]) followed.

### 2.4. Infectivity Assays

RVA was titrated in MA104 cells prepared in MM containing 1 μg/mL of trypsin using 12-well cell culture plates for plaque assay as described previously [29]. BCoV was titrated by TCID_50_/mL, based on assay by Spearman–Karber method [30] in HRT-18 cells in MM containing 1 μg/mL of trypsin using 96-well cell-culture plate at three dpi.

The reduction factor (RF) was used for determining the viral inactivation. Viral inactivation was considered effective when the RF was greater than or equal to 3 log_10_ [31].

## 3. Results

### 3.1. Suspension Test for Evaluating the Virucidal Activities of the Solutions against Viruses in the Aqueous Phase

As shown in Table 1, 0.17% FdCa(OH)_2_ solution and Mix500 could inactivate RVA to the effective level within 5 s, even in the presence of 5% FBS. Mix500 inactivated RVA to undetectable levels (<1.69 ± 0.00 PFU/mL; RF > 5.13) within 5 s, while QACx500 could not inactivate RVA to the effective level in 1 min. By contrast, QACx500, 0.17% FdCa(OH)_2_ solution and Mix500 were found to reduce BCoV virus titer to the effective level within 5 s in the presence of 5% FBS. When the blocking solution was added before the addition of the virus (0 s), the viral titers were similar to the positive control. Thus, the blocking solution evidently stopped the virucidal activity of the tested solutions, when mixed in equal amounts with the reaction solution.

### 3.2. Evaluation of Virucidal Activities of Mix500 toward Viruses on the Contaminated Rayon Sheets by the Dipping Techniques

The capacity of Mix500 to inactivate RVA and BCoV in the presence of 5% FBS on rayon sheets through dipping technique is shown in Table 2. Mix500 could inactivate RVA up to undetectable levels (<1.69 ± 0.00 PFU/mL; RF > 4.11) within 30 s, and yet inactivated BCoV up to undetectable levels (<2.50 ± 0.00 TCID_50_/mL; RF > 3.66) within 15 s. When the blocking solution was added before the addition of the virus (0 s), the viral titers were similar to the positive controls.

### 3.3. Evaluation of Virucidal Activities of Mix500 toward Viruses on the Contaminated Rayon Sheets by the Dropping Techniques

In contrast to the results of the dipping technique, the dropping techniques with Mix500 could not reduce the viral titers on the rayon sheets to the effective level even after 1 h for RVA and 10 min for BCoV, as shown in Table 2.

### 3.4. Evaluation of Virucidal Activities of Viruses on Rubber Surfaces Contaminated with Feces

The recovery ratio of the viruses with dW_2_ from the surface of rubber carrier after drying for 1 h was measured as 40% for RVA and 1% for BCoV, respectively (Table 3). The dropped Mix500 achieved a three log_10_ reduction of RVA containing 5% feces on the rubber carrier within 5 min. The dropped Mix500 inactivated RVA to an undetectable level (<1.69 ± 0.00 PFU/mL; RF > 3.55) within 10 min. On the other hand, BCoV on the rubber carrier was inactivated to the effective level within 1 min by the drop of Mix500 (Table 3).

## 4. Discussion

The importance of a cleaning or washing step for proper disinfection is well known, and this was confirmed here. In the present study, two enteric bovine viruses, classified according to the presence or absence of an envelope, were used to evaluate the disinfectants by suspension and carrier tests.

In the suspension tests, QACx500 could not inactivate RVA in 1 min. It is indeed reasonable that a non-enveloped virus will not be inactivated with QAC [32,33]. FdCa(OH)_2_ was shown to be able to inactivate BCoV, an enveloped virus, in 5 s, as well as an avian coronavirus, i.e., infectious bronchitis virus (IBV) [4], in contrast to avian influenza virus (AIV) or Newcastle disease virus (NDV), which required more than 10 min [17]. These data suggest that coronaviruses are susceptible to alkalinity such as pH 12.4.

The spectral broadening effect of QAC in aqueous phase was observed toward RVA within 5 s at Mix500. As shown in Table 1, QACx500 failed to inactivate RVA in the suspension test, as well as infectious bursal disease virus (IBDV) (non-enveloped virus) as described [17]. Mix500 has a broad spectrum virucidal effect, resulting in a more effective virucidal agent, with a wide spectrum of pathogen inactivation [4,16,17]. Therefore, Mix500, the mixture of QAC and FdCa(OH)_2_, was selected for the carrier tests.

People and vehicles can be important pathways, mechanically, for the introduction of new diseases into farms. Fomites carried by people (boots, clothes, etc.) or even the people themselves, through contaminated skin, can spread various pathogens onto farms [34]. Compared to pathogens in the aqueous phase, pathogens on abiotic carriers or in feces have long been demonstrated to be more resistant to disinfectants [1,14,18]. It has been suggested that carrier tests along with suspension tests, which simulate practical conditions, are required to evaluate disinfectants [35,36]. Otherwise, it may not be possible to make appropriate deductions and decisions concerning the effectiveness and usability of disinfectants toward pathogens deposited on fomites and surfaces.

Generally, enteric viruses are highly resistant, especially under environmental conditions. The viruses can survive on the surface of materials, fomites and food for long periods [8,36]. Therefore, our model involved rayon sheets as equivalents to clothes that might be contaminated with RVA and BCoV, which are found in organic materials such as FBS. As shown in Table 2, with complete dipping techniques, Mix500 could inactivate RVA and BCoV within 30 s and 15 s, respectively. Furthermore, the dropping method to mimic spraying disinfectant on coveralls or clothes of farm staffs was ineffective against RVA and BCoV on the rayon sheets, even after 1 h and 10 min, respectively (Table 2). Normally, at bovine farms, showering-in is not popular. Clothes hygiene for farm personnel is an important aspect of human management. It is recommended that used clothes be replaced daily with clean ones.

In the present study, the rubber coupon was used for simulating farm personnel boots, and using the dropping method instead of footbath showed the effect of Mix500 toward RVA and BCoV within 10 and 1 min, respectively as shown in Table 3. In the rubber carrier test, the stabilities of the two test viruses, RVA and BCoV, were examined after drying process and the dropping technique. Miyaoka et al. found that the survival rate of viruses on plastic carriers was higher for IBV than for AIV [28]. Our results showed that the recovery ratio of RVA from rubber carriers was higher than that of BCoV (Table 3). Thus, the drying process in the dropping technique reduced the titer of viruses, but RVA may be more stable in the drying process than BCoV.

In the presence of organic matter such as litter and feces, disinfectants required more time to inactivate viruses and became less effective, suggesting that a cleaning step is important before disinfection [1]. In our previous reports, it was demonstrated that changing boots at the entrance of each calf shed and performing appropriate foot baths after boot use inside the shed results in lowered virus detection rates and lowered mortality rates [19,21]. These results suggested that changing boots and clothes is necessary for preventing pathogens from entering the hygiene management area. In order to achieve this, clothing and footwear or boots must be properly disinfected after contact with feces or mud.

Demonstrating an accurate evaluation system for appraising disinfectants is very critical to prevent misleading results. The data in Table 1, Table 2 and Table 3 show a very considerable gap between the suspension and carrier tests. If only the suspension tests are used, all disinfectants seem to be effective. In the carrier tests, differences were also observed between dipping and dropping methods. It is hence suggested that different evaluation systems should be used to identify differences in disinfectant properties for actual use in livestock production settings. Accurate evaluation systems of disinfectants would minimize such severe errors during application of disinfectants on livestock farms.

It is important to select disinfectants that are effective against a wide spectrum of pathogens under the conditions normally found on livestock farms. Mix500 can be applied as a disinfectant or a microbiocidal agent which can inactivate enveloped and non-enveloped viruses and bacteria in contaminated fomites, especially in mammals and poultry farms [1,4,16,17]. In Japan, the Ministry of Agriculture, Forestry and Fisheries (MAFF) recommends farmers to use agricultural slaked lime with a Ca(OH)_2_ content of 65–75% for the enforcement of biosecurity measures; however, slaked lime failed to show synergistic effects with QAC toward AIV and NDV at low temperatures [16]. FdCa(OH)_2_ seems to be much safer and more effective than slaked lime, since the former has a much smaller particle size (average 10 µm) and is purer (97% Ca(OH)_2_) than slaked lime (less than 150 µm). FdCa(OH)_2_ would not damage or bleach boots, clothes and floors. These characteristics are superior to slaked lime or sodium hypochlorite (NaOCl). The findings of the present study can also help farmers to properly apply FdCa(OH)_2_ with QAC-based disinfectants in appropriate concentrations and exposure times in farms, in order to prevent and control infectious disease outbreaks.

Routine disinfection in livestock farms is an important hygienic measure to prevent the transmission of pathogens from a previous flock/shed to the next production cycle [3,37]. It is possible to increase the level of biosecurity at bovine farms through the farm HACCP approach by changing boots and clothing, the appropriate use of disinfectants, or limiting access to the farms. In such cases, our findings would help livestock farmers implement better infection control strategies.

## Figures and Tables

**Figure 1 microorganisms-10-01320-f001:**
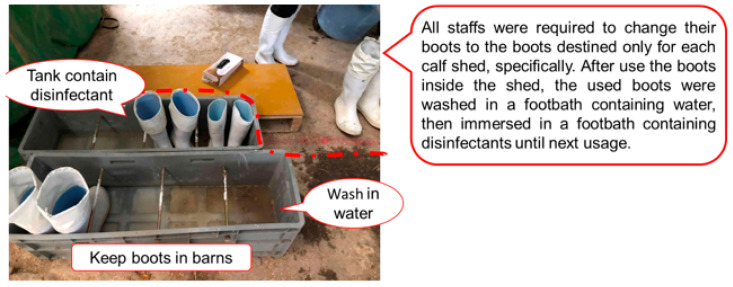
Biosecurity measures strengthened through boot hygiene at Farm A. Disinfectants: QAC (Rontect^®^, Scientific Feed Laboratory Co., Ltd., Tokyo, Japan), diluted 1:500 with tap water and added FdCa(OH)_2_ powder (Fine Co., Ltd., Tokyo, Japan) at the final concentration of 0.17% at the exit of each calf shed.

**Figure 2 microorganisms-10-01320-f002:**
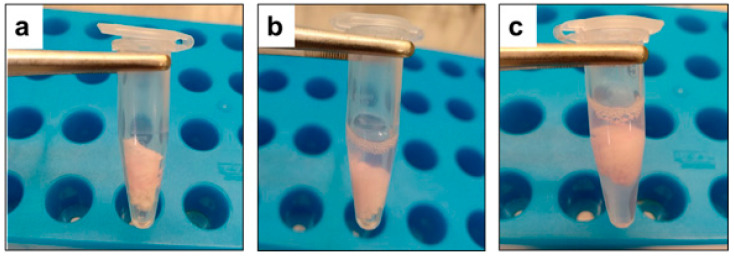
(**a**) One hundred microliter of virus containing 5% FBS was spotted on a piece of rayon sheet (double-fold, size 2.0 cm × 2.0 cm) in 1.5 milliliters microtube and air-dried for 30 min inside the biosafety cabinet at room temperature. (**b**) Four hundred microliters of Mix500 test solution was used for treatment onto the contaminated dried surfaces through complete dipping. (**c**) Virus inactivation was blocked by adding five hundred microliters of the blocking solution.

**Figure 3 microorganisms-10-01320-f003:**
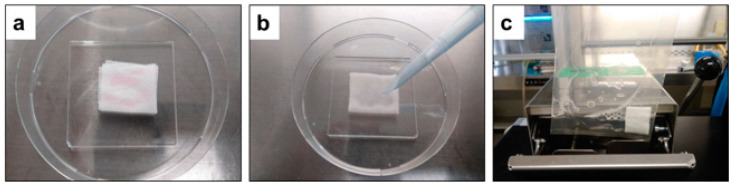
(**a**) One hundred microliters of virus containing 5% FBS was spotted on a piece of rayon sheet (four-fold, size 2.5 cm × 2.5 cm) and air-dried for 30 min inside the biosafety cabinet at room temperature. (**b**) Five hundred microliters of Mix500 test solution was dropped on rayon carrier to cover all spotted area for treatment onto the contaminated dried surfaces. (**c**) Virus inactivation was blocked by placing the rayon sheet into stomacher bags containing two milliliters of the blocking solution with stomaching.

**Figure 4 microorganisms-10-01320-f004:**
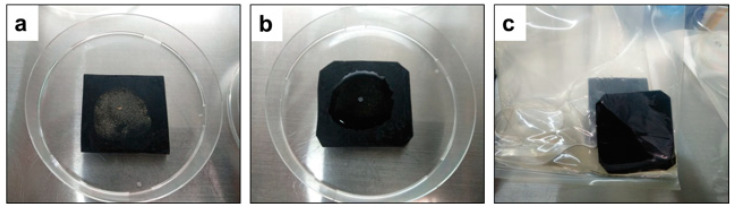
(**a**) One hundred microliters of virus containing 5% calf feces was spotted on a rubber carrier coupon (around 5.0 cm × 5.0 cm) and subsequently spread by sterile glass spreader onto the carriers and air-dried for 60 min inside the biosafety cabinet at room temperature. (**b**) Five hundred microliters of Mix500 test solution used for treatment onto the contaminated dried surfaces. (**c**) Virus inactivation was blocked by placing the rubber carrier into stomacher bags containing two milliliters of the blocking solution.

**Table 1 microorganisms-10-01320-t001:** Virucidal efficacy of the tested solutions toward bovine rotavirus A (RVA) and bovine coronavirus (BCoV) in aqueous phase containing 5% fetal bovine serum.

Tested Solutions	Virus	t_pc_ ^(a)^	0 s ^(b)^	5 s ^(c)^	1 min
QACx500	RVA	6.82 ± 0.27	6.76 ± 0.28	5.91 ± 0.00	5.53 ± 0.07
FdCa(OH)_2_			6.15 ± 0.04	2.15 ± 0.19 *	NT
Mix500			5.98 ± 0.03	<1.69 ± 0.00 *	NT
QACx500	BCoV	6.91 ± 0.06	6.75 ± 0.00	<2.50 ± 0.00 *	NT
FdCa(OH)_2_			6.66 ± 0.06	<1.50 ± 0.00 *	NT
Mix500			6.66 ± 0.06	<2.50 ± 0.00 *	NT

^(a)^ The titer converted into an index in log_10_ of virus control, for RVA (log_10_PFU/mL) and for BCoV log_10_TCID_50_/mL, respectively. ^(b)^ Blocking solution added before addition of viruses. ^(c)^ The titer converted into an index in log_10_ of the recovered virus after indicated duration of treatment such as 5 s and 1 min. NT: Not tested. * Inactivation regarded as effective when RF was greater than or equal to 3.

**Table 2 microorganisms-10-01320-t002:** Virucidal efficacy of Mix500 toward bovine rotavirus A (RVA) and bovine coronavirus (BCoV) in rayon sheet carrier.

Incubation Time		RVA		BCoV	
		Virus Titer (log_10_PFU/mL)	RF ^(d)^	Virus Titer (log_10_TCID_50_/mL)	RF ^(d)^
t_pc_ ^(a)^		5.81 ± 0.01		6.16 ± 0.06	
0 s ^(b)^	Dipping-carrier	5.17 ± 0.04	0.63	5.58 ± 0.06	0.58
5 s ^(c)^		4.54 ± 0.01	1.26	3.25 ± 0.00	2.91
15 s		3.11 ± 0.03	2.69	<2.50 ± 0.00 *	>3.66
30 s		<1.69 ± 0.00 *	>4.11	NT	NT
t_pc_ ^(a)^		5.82 ± 0.02		6.16 ± 0.06	
10 min ^(c)^	Dropping-carrier	5.56 ± 0.01	0.25	3.25 ± 0.00	2.91
1 h		4.59 ± 0.00	1.22	NT	NT

^(a)^ The titer converted into an index in log_10_ of virus control. ^(b)^ Blocking solution added before addition of viruses. ^(c)^ The titer converted into an index in log_10_ of the recovered virus after indicated duration of treatment such as 0 s, 5 s, 15 s, 30 s, 10 min and 1 h. ^(d)^ Reduction factor = log_10_ (titer of control/mL)—log_10_ (titer of treated samples/mL). NT: Not tested. * Inactivation regarded as effective when RF was greater than or equal to 3.

**Table 3 microorganisms-10-01320-t003:** Virucidal efficacy of the Mix500 toward bovine rotavirus A (RVA) and bovine coronavirus (BCoV) in rubber carrier.

Incubation Time	RVA		BCoV	
	Virus Titer (log_10_PFU/mL)	RF ^(c)^	Virus Titer (log_10_TCID_50_/mL)	RF ^(c)^
t_pc_ ^(a)^	5.25 ± 0.01		4.50 ± 0.00	
15 s ^(b)^	4.05 ± 0.04	1.19	3.58 ± 0.06	0.91
30 s	3.95 ± 0.05	1.30	2.66 ± 0.06	1.83
1 min	3.50 ± 0.06	1.74	<1.50 ± 0.00 *	>3.00
3 min	3.45 ± 0.02	1.80	NT	NT
5 min	1.89 ± 0.08 *	3.35	NT	NT
10 min	<1.69 ± 0.00 *	>3.55	NT	NT

^(a)^ The titer converted into an index in log_10_ of virus control. ^(b)^ The titer converted into an index in log_10_ of the recovered virus after indicated duration of treatment, i.e., 15 s, 30 s, 1 min, 3 min, 5 min and 10 min. ^(c)^ Reduction factor=log_10_ (titer of control/mL)—log_10_ (titer of treated samples/mL). NT: Not tested. * Inactivation regarded as effective when RF was greater than or equal to 3.

## Data Availability

All relevant data are presented in the article. The data that support the findings of this study are available from the author M.A.H. upon reasonable request.

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
