# Peer review of "Evaluation of Virucidal Quantitative Carrier Test towards Bovine Viruses for Surface Disinfectants While Simulating Practical Usage on Livestock Farms"

_microorganisms, 2022, doi:10.3390/microorganisms10071320_

Round 1

Reviewer 1 Report

The current manuscript entitled "Evaluation of virucidal quantitative carrier test towards bovine viruses for surface disinfectants while simulating practical usage on livestock farms " Hasan et al., (2022), demonstrated a properly apply of FdCa(OH)2 with QAC based disinfectants in appropriate concentrations and exposure times in farms, in order to prevent and control infectious disease outbreaks. The manuscript language is well written.

Major comments:

11-   The results are not described well, it should be linked with the figures in the end of manuscript.

22- The formatting of manuscript needs to obey the rules of Microorganisms journal.

33- The manuscript needs more English revision by native English person.

Author Response

Thank you very much for reviewing our manuscript. All comments are helpful to upgrade the manuscript and make it better to understand by readers. We want to revise the manuscript according to your comments. We presented our answers here, as well as the corrected parts in the manuscript, in yellow highlights. The “Track Changes” function was not used because figures and tables were inserted in the text, complicating the tracking of change history.

Sincerely yours,

Kazuaki Takehara,

Reviewer 2 Report

Interesting article.

I have comments to be discussed in the article:

1. Describe the synergism of Quaternary ammonium compounds and alkali compounds

2. ASTM methods are used to assess virucidal activity in the USA, in Europe, for example, EN 14675 , EN 17122.Please write about these methods.

3. What standards were used in the study?

4.What was Quaternary ammonium compounds used in the research?

Author Response

Reply to the Reviewer comments.

Dear Editor and reviewers,                                                                                   28, June, 2022

Thank you very much for reviewing our manuscript. All comments are helpful to upgrade the manuscript and make it better to understand by readers. We want to revise the manuscript according to your comments. We presented our answers here, as well as the corrected parts in the manuscript, in yellow highlights. The “Track Changes” function was not used because figures and tables were inserted in the text, complicating the tracking of change history.

Sincerely yours,

Kazuaki Takehara,

Round 2

Reviewer 1 Report

Thanks for amendment.